# Application of Microbiome Management in Therapy for *Clostridioides difficile* Infections: From Fecal Microbiota Transplantation to Probiotics to Microbiota-Preserving Antimicrobial Agents

**DOI:** 10.3390/pathogens10060649

**Published:** 2021-05-24

**Authors:** Chun-Wei Chiu, Pei-Jane Tsai, Ching-Chi Lee, Wen-Chien Ko, Yuan-Pin Hung

**Affiliations:** 1Department of Internal Medicine, Tainan Hospital, Ministry of Health and Welfare, Tainan 700, Taiwan; bahamudo@gmail.com; 2Department of Medical Laboratory Science and Biotechnology, National Cheng Kung University, Medical College, Tainan 704, Taiwan; peijtsai@mail.ncku.edu.tw; 3Clinical Medicine Research Center, National Cheng Kung University Hospital, College of Medicine, National Cheng Kung University, Tainan 704, Taiwan; chichingbm85@gmail.com; 4Department of Internal Medicine, College of Medicine, National Cheng Kung University Hospital, Tainan 704, Taiwan; 5Department of Medicine, College of Medicine, National Cheng Kung University, Tainan 704, Taiwan

**Keywords:** *Clostridioides difficile*, *Clostridioides difficile* infection, microbiome, probiotics, recurrence, fecal microbiota transplantation

## Abstract

Oral vancomycin and metronidazole, though they are the therapeutic choice for *Clostridioides difficile* infections (CDIs), also markedly disturb microbiota, leading to a prolonged loss of colonization resistance to *C. difficile* after therapy; as a result, their use is associated with a high treatment failure rate and high recurrent rate. An alternative for CDIs therapy contains the delivery of beneficial (probiotic) microorganisms into the intestinal tract to restore the microbial balance. Recently, mixture regimens containing *Lactobacillus* species, *Saccharomyces boulardii*, or *Clostridium butyricum* have been extensively studied for the prophylaxis of CDIs. Fecal microbiota transplantation (FMT), the transfer of (processed) fecal material from healthy donors to patients for treating CDIs, combined with vancomycin was recommended as the primary therapy for multiple recurrent CDIs (rCDIs). Either probiotics or FMT have been utilized extensively in preventing or treating CDIs, aiming at less disturbance in the microbiota to prevent rCDIs after therapy cessation. Otherwise, many newly developed therapeutic agents have been developed and aim to preserve microbiota during CDI treatment to prevent disease recurrence and might be useful in clinical patients with rCDIs in the future.

## 1. Disruption of Gut Microbiota after Antibiotic Exposure Results in Recurrent *C. difficile* Infection

*Clostridioides difficile,* as the major cause of antibiotic-associated diarrhea, has clinical symptoms ranging from diarrhea to pseudomembranous colitis or toxic megacolon, with a mortality rate of up to 25–40% [1,2,3,4,5]. In the “Clinical Practice Guidelines for *Clostridioides difficile* Infections (CDIs) in Adults and Children: 2017 Update” by the Infectious Diseases Society of America (IDSA) and Society for Healthcare Epidemiology of America (SHEA), vancomycin replaced metronidazole as the therapeutic choice for either mild-to-moderate or severe CDIs [6]. Nevertheless, oral vancomycin and metronidazole markedly disturbed the microbiota, which resulted in the dense colonization by vancomycin-resistant Enterococcus, *Klebsiella pneumoniae,* and *Escherichia coli*, and, more importantly, the prolonged loss of colonization resistance to *C. difficile* [7]. Although antimicrobial resistance is not clinically problematic, treating CDIs with metronidazole and vancomycin is associated with a high treatment failure rate and recurrence rate [8]. For recurrent CDIs (rCDIs), according to the Clinical Practice Guidelines by IDSA/SHEA, oral vancomycin is still the drug of choice, but the rate of the sequential recurrence is as high as 22.6–41.8% despite successful treatment with vancomycin [9].

There are more than 1000 distinct species of bacteria inhabited in the human gastrointestinal system with a symbiotic relationship, with the collection of microbes called the “microbiota” [10,11]. The relative high susceptibility to infections in neonates might be due to the immaturity of the immune system, and the mechanism is probably due to the relative “immature” microbiota [12]. By colonizing adult germ-free mice with the cecal contents of neonatal mice, the microbiota is unable to prevent colonization by two bacterial pathogens, including *Citrobacter rodentium*, a natural pathogen of mice that is used to model human infections with the enteropathogenic *E. coli* and *Salmonella typhimurium* that cause mortality in neonates [12]. The lack of colonization resistance was correlated to the absence of Clostridiales in the neonatal microbiota, and the administration of Clostridiales could protect neonatal mice from pathogen infection [12]. Thus, the component of gut microbiota is associated with the immunity of the host.

The colonized microbiota in the mammalian gut occurs shortly after birth and remains with little fluctuation throughout the host’s life; the microbiota is primarily composed of five bacteria phyla in healthy adults: Firmicutes (79.4%), Bacteroidetes (16.9%), Actinobacteria (2.5%), Proteobacteria (1%), and Verrucomicrobia (0.1%) [13]. A wide range of host or environment factors, including diet, sleep, and disease, can alter the microbiota diversity and abundancy [10,11]. Of note, the rise of antimicrobial agent-resistant pathogens, combined with reduction of microbiota diversity after antibiotic treatment, has become a significant challenge in the fight against all kinds of invasive infections worldwide [14]. To prevent the disruption of the microbiota, some microbiota-based treatments, such as fecal microbiota transplantation (FMT) and the administration of probiotics, have been used to “rescue” the disrupted microbiota [14] (Figure 1).

FMT, the transfer of (processed) fecal material from healthy donors to CDI patients, combined with vancomycin have been recommended as the primary therapy for multiple rCDIs [15] (Table 1). To simplify the influence of FMT on gut microbiota, studies among specific populations, such as groups with malignancy or inflammatory bowel disease, are not discussed in this review [16]. Since the poor bacterial diversity is correlated with rCDIs in clinical patients, the efficacy of FMT in preventing rCDIs has been correlated with the restoration of the disturbed and poor diversity of gut microbiota due to the disruption by antibiotic exposures [17,18], for example, replanting the genera of Bacteroidetes [16,19,20,21,22,23,24,25], Firmicutes [23,25], *Faecalibacterium* [16,26], or *Bifidobacterium* [27], and conveniently decreasing Proteobacteria [22,25,28], Enterobacteriaceae [29,30], or bacteria harboring antibiotic-resistant genes within the microbiota [25].

The diverse change in microbiota after FMT was noticed, which might depend on the different composition of microbiota of the donor population, different donor ages, or different FMT methods [22,23,27,31,32]. For all successful FMT treatments with the resolution of rCDI symptoms and a negative *C. difficile* toxin test within 4–12 weeks after FMT, the genomic analysis of donor microbiota showed that the Bacteroidetes-to-Fermicutes ratio did not reveal a significant difference among donors with different ages [32]. However, the relative abundance of phylum Actinobacteria and family Bifidobacteriaceae was notably reduced in donors of more than 60 years of age [32]. A FMT study that consisted of the elderly with rCDIs revealed that FMT resulted in a marked improvement in all clinical parameters and overall microbiota diversity, but this response was less vigorous than the younger group [28]. Additionally, in the microbiota analysis, Firmicutes did not change markedly, but Proteobacteria decreased significantly in post-FMT samples among the elderly patients experiencing rCDIs [28].

There are numerous ways of performing FMT, including nasojejunal tube [21,22,30], colonoscopy [18,20,21,22,25,30,33] and encapsulated oral form [19,20,23]. The way of performing FMT might affect the microbiota distribution [20]. In a randomized trial of adults with ≥3 episodes of rCDIs who received encapsulated lyophilized fecal microbiota or frozen FMT by enema, the rCDI was prevented equally among the capsule group and FMT enema group (84% vs. 88%, respectively, *p* = 0.76). Although both products notably normalized the diversity of fecal microbiota, the lyophilized orally administered product was less effective in replanting Bacteroidia and Verrucomicrobia classes, compared to the frozen product via enema, and it was likely that there were some damaging effects of gastric acid, bile salts, and digestive enzymes during the upper gastrointestinal transit for the orally administered FMT product [20].

The colonization of Bacteroidetes [16,19,20,21,22,23,24,25], Firmicutes [23,25], *Faecalibacterium* [16,26], or *Bifidobacterium* [27] were found to be decreased during the initial CDIs and rCDIs after successful FMT. In a cohort with a long-term (up to 409 days) follow-up, all patients who were clinically recovered and free of CDIs were characterized by increased members of the genera *Bacteroides, Parabacteroides,* and *Faecalibacterium* throughout the year in their fecal microbiota [19]. In a prospective study dealing with rCDI children, FMT successfully prevented rCDI episodes for at least 3 months, along with significantly increased levels of Bacteroidetes [21]. Among rCDI patients, FMT reduced beta diversity differences between the donors and recipients and increased in relative abundance of *F. prausnitzii* [16]. Of patients receiving FMT for rCDIs, symptoms resolved in 71.4% of cases treated with the fecal bacterial composition dominated by Firmicutes, Bacteroidetes, and Proteobacteria, and were remarkably stable over time after FMT [22]. With encapsulated oral intake FMT, taxa within the Firmicutes showed rapid increases in relative abundance that did not vary significantly over time. Bacteroidetes taxa only showed significant increases in abundance after one-month post-FMT among patients with rapid decline in the rCDI symptoms [23]. Among patients receiving FMT using colonoscopy in Germany, the healing rate of CDIs was 94%, and in all patients successfully treated with FMT, their microbiota revealed elevated Lactobacillaceae, Ruminococcaceae, Desulfovibrionaceae, Sutterellaceae, and Porphyromonodacea [33]. In a high-throughput microbiota profiling using a phylogenetic microarray analysis for rCDI patients, FMT reverted the patients’ bacterial community to become dominated by *Clostridium* clusters IV and XIVa and there was an increase in Bacteroidetes [24].

No matter the composition of donor’s microbiota or the ways of performing FMT, donor-derived Bacteroidetes, Firmicutes, or *Bifidobacterium* can colonize rCDI patients who were treated successfully with FMT for more than one year, and accordingly, FMT may have long-term consequences for the recipient’s microbiota and health [22,23,24,27,31].

Proteobacteria [22,25,28], Enterobacteriaceae [29,30] and the bacteria harboring antibiotic resistance genes within the microbiota [25] were mostly increased during CDIs and decreased after FMT. In a prospective, observational study of rCDI children, FMT successfully prevented rCDIs for a minimum of 3 months post-FMT, with no major adverse effects, along with the significantly decreased level of Proteobacteria [21].

Some commercialized products have been developed for FMT [34]. To utilize FMT with a high abundance of non-resistant species to displace antibiotic-resistant organisms from the recipient’s microbiome, RBX2660, a liquid suspension of donor microbiota, has recently been deployed to treat rCDIs [34]. RBX2660 was found to aid in the successful prevention of rCDIs, correlated with the taxonomic convergence of patient microbiota to the donor microbiota, and also dramatically reduced the abundance of antibiotic-resistant Enterobacteriaceae [34].

The mechanisms of FMT to effectively prevent or treat CDIs by modulating microbiota were proposed to be through the influence on the metabolism of certain bile acids that affect germination or vegetative growth of *C. difficile* [31,35,36,37,38]. Patients with CDIs exhibited significantly lower fecal levels of secondary bile acids and higher levels of primary bile acids [31]. In sum, the FMT mechanism might include microbiota-associated bile salt hydrolases (BSHs) [35], the bile acid-farnesoid X receptor-fibroblast growth factor pathway [36], and bile acid-inducible (bai) operon [37]. Loss of gut microbiota-derived BSHs predisposes individuals to CDIs by perturbing the gut bile metabolism, and the restoration of gut BSH functionality has contributed to efficacies of FMT in treating rCDIs [35]. FMT for rCDIs is accompanied by a significant, sustained increase in circulating levels of FGF19 and in the reduction of FGF21, which are critical pathway signals of the bile acid-farnesoid X receptor-fibroblast growth factor, which are important pathways in the restoration of gut microbiota and bile acid profiles [36]. A single bai operon (baiCD), majorly found in *C. scindens* and *C. hiranonis*, was recently reported to protect against *C. difficile* colonization, and is a required gene for 7α-dehydroxylation, which is a key step in the transformation of primary to secondary bile acids. The rCDI patients were baiCD-negative at baseline, but baiCD turned positive after successful FMT from a baiCD-positive donor [37].

Other than bile acid, sustained increases in the levels of the short chain fatty acids (SCFAs), including butyrate, acetate, and propionate were observed in FMT recipients, and these metabolites that increased following FMT were associated with the repletion of bacteria classified within the Lachnospiraceae, Ruminococcaceae, and unclassified *Clostridial* families [38].

Some treatment failures were reported in FMT for CDI. In a study that prospectively treated recurrent CDI with FMT by colonoscopy in Italy, only 44 (69%) were cured by a single fecal infusion, whereas the other 20 (31%) needed repeat infusions [39]. At multivariate analysis, severe CDI (OR 24.66) and inadequate bowel preparation (OR 11.53) were found to be independent predictors of failure after a single fecal infusion [39]. In the United States, the primary cure rate of FMT at 3 months in a multicenter study toward CDI was as low as 58.7% in solid organ transplant recipients, which was lower in patients with other co-morbidity, including 25% of patients with underlying inflammatory bowel disease who had worsening disease activity, while 14% of cytomegalovirus-seropositive patients had reactivation [40]. Predictors of failing for a single FMT included inpatient status, severe and fulminant CDI, presence of pseudomembranous colitis, and use of non-CDI antibiotics at the time of FMT [40]. Patients with high risk for FMT failure should be monitored closely and might need repeat FMT.

There were some adverse events related to FMT that should be taken into considerations, including nausea, diarrhea, bloating, and abdominal cramping [10]. Of note, two patients with the same stool donor experienced *E. coli* bacteriemia, which lead to the death of one of the patients [41]. Thus, the selection of appropriate fecal donors and recipients is an important issue in performing FMT.

## 2. Probiotics Supply to Restore the Disturbed Microbiota in CDI

An alternative for CDI therapy comprises the delivery of beneficial (probiotic) organisms into the intestinal tract to restore the microbial balance. The theoretical foundation for this approach is that specific components of the microbiota, the microorganisms that normally colonize the body, can protect the gut from some pathogenic bacteria. Nevertheless, during antibiotic treatment for other infectious diseases, the protective intestinal microflora is damaged, which leads to the *C. difficile* proliferation and infection. The initial antibiotic exposure leaves the host susceptible to colonization and subsequent infection by *C. difficile*. A so-called “second-hit” to the intestinal microbiota occurs when the infected host is treated with metronidazole or vancomycin, further destroying susceptible microbiota [42]. Probiotic microorganisms, such as *Lactobacillus rhamnosus* GG. or *Saccharomyces boulardii*, have been studied for the prophylaxis or treatment of CDIs with moderate certainty evidence in the meta-analysis [43].

*Bifidobacterium* are a genus of Gram-positive, nonmotile, often branched anaerobic bacteria as ubiquitous residents of the gastrointestinal tract [44]. Supernatants from *bifidobacteria* isolated from healthy infants had the ability to inhibit *C. difficile* growth and adhesion to enterocytes [45]. Consumption of *B. bifidum* modulates the dominant intestinal bacterial taxa in healthy adults [46]. The efficacy of a mixture of *Lactobacillus* species in treating or preventing CDIs has been proven in many clinical trials [47,48,49,50]. A four-strain oral probiotic capsule containing *L. acidophilus* NCFM, *L. paracasei* Lpc-37, *B. lactis* Bl-07, and *B. lactis* Bl-04 has been used as the adjunct probiotic with significant improvement in diarrhea outcomes in patients with CDIs [47]. The consumption of *L. acidophilus* (CL1285), *L. casei* (LBC80R), and *L. rhamnosus* CLR2 combination (Bio-K+) was reported to diminish the CDI incidence and remained at low mean levels of 2.3 cases per 10,000 patient days among adult inpatients treated with antibiotics [48]. *L. plantarum* 299v could reduce the CDI incidence from 1.21% to 0.11% among hospitalized patients with comorbid nephrology or transplantation disease [49]; *L. casei*, *L. bulgaricus*, and *Streptococcus thermophiles* could prevent the CDI with an absolute risk reduction of 17% (7–27%) among hospital patients taking antibiotics [50]. Taken jointly, the evidence from these clinical trials suggests that the *Lactobacillus* species serves as an effective and safe probiotic in preventing CDI or its recurrence, but the role of *Lactobacillus* species as adjuvant therapy for CDIs remains unclear [47,48,49,50].

*S. boulardii,* as one kind of yeast probiotics, was characterized by its minimum effect on normal microbiota in healthy humans, whereas, in contrast, it could restore antibiotic-disrupted microbiota rapidly [51]. *S. boulardii* CNCM I-745 significantly reduced cecal tissue damage, NF-κB phosphorylation, and TNFα protein expression caused by CDIs in the hamster model [52]. Among patients with CDIs who received high-dose vancomycin therapy, *S. boulardii* treatment could reduce the recurrence rate of CDIs [53]. In another CDI study, treatment with *S. boulardii* significantly reduced its recurrence risk (relative risk, 0.43) [54]. Currently, *S. boulardii* is the most commonly used of the probiotic mixture regimens for the CDI prophylaxis [43].

Some of the *Bacteroides* genus have been found to be capable of modulating microbiota, inhibiting *C. difficile*, and thus serving as surrogates for probiotics for antibiotic-associated diarrhea [55,56]. *B. fragilis* strain ZY-312 could increase the species *Akkermansia muciniphila*, which is an important human intestinal mucin-degrading bacterium in the microbiota and against CDIs, possibly by resisting its colonization and improving the integrity and function of the gut barrier [56]. *B. thetaiotaomicron* could significantly reverse the decreased Bacteroidetes levels as well as the increased Proteobacteria and Verrucomicrobiota levels, thus suppressing *C. difficile* in the gut [55]. The safety of using *Bacteroides* genus as surrogates for probiotics for antibiotic-associated diarrhea warrants further evaluation, since many *Bacteroides* genus could be invading pathogens in clinical patients [55,56].

*C. butyricum*, facultative, and strictly anaerobic bacteria are progressively colonized in the neonate’s intestine following birth [57]. In the rat model, the cytotoxin titer of *C. difficile* in rat feces decreased after treatment with *C. butyricum* [58]. *C. butyricum,* compared to the placebo, decreased the rate of antibiotic-associated diarrhea in pediatric patients from 59% to 5%, which was associated with an increase in the level of anaerobes and prevented the decrease in the level of *Bifidobacterium* species [59]. *C. butyricum,* combined with *S. faecalis* and *Bacillus mesentericus,* prevented CDIs with a crude odds ratio of 0.074 among elderly patients who underwent proximal femoral fracture surgery [60]. The evidence of *C. butyricum* as an adjuvant therapeutic choice for CDIs is still lacking.

Recently, the mixture regimens containing *Lactobacillus* species, *S. boulardii*, or *C. butyricum* have been extensively studied for CDI prophylaxis [43,60]. The efficacy of the probiotic mixture for treating or preventing CDIs has been demonstrated, which was supposed to be through the modulation of gut microbiota; although, the influence of the single component of mixture regimens on microbiota was illustrated, the overall effect of the probiotic mixture in gut microbiota was not demonstrated [43,47,48,49,50,60].

There were no head-to head randomized clinical trials to compare the efficacy of two different probiotics. Thus, it is difficult to tell if one kind of probiotics are superior to another ones. Nevertheless, multi-strain probiotics (mixture regimen) might be more useful than single-strain options because of synergy and additive effects among the individuals with CDI in modulation of the immune system and gut microbiota [43,60].

Some debates exist in the use of probiotics. First, the fed probiotic bacteria could be detected in stool samples of all participants when consuming the probiotics; however, the probiotic bacteria was found in the colonic mucosa in only some participants [61,62]. The transient engraftment was dependent on the microbiome composition of the participants, which was quite different from the more uniform results obtained from the study of germ-free mice [61,62]. Selecting appropriate probiotics on the basis of the composition of the microbiota of the recipient, the “target therapy” might be the goal of development in the future.

Though the beneficial effect of probiotics in preventing CDI has been illustrated, there are still some concerns regarding their safety, including infections or inflammatory/ fatal effects derived from toxins produced either by the probiotic strains or by possible bacterial contaminants [10]. Of note, *Lactobacillus* infection after taking probiotics products contain *Lactobacillus* spp. had been reported in some relatively immunocompromised patients [63,64,65]. These immunocompromised patients were at higher risk of developing CDI, and the use of probiotics to prevent CDI in these group of patients should be approached more carefully.

## 3. Therapeutic Agents Preserving Microbiota during *C. difficile* Infection Treatment

Many newly developed therapeutic agents are aimed at preserving microbiota during CDI treatment to prevent disease recurrence; for this purpose, two major kinds of therapeutic strategy were investigated: first, the development of narrow-spectrum antimicrobial agents, such as fidaxomicin [66,67,68,69], ridinidazole [70,71], or cadazolid [72,73]; second, developing antimicrobial agents specifically targeting some distinct structure of *C. difficile*, such as anti-sense antimicrobial agents [74,75], anti-toxin antibodies [76,77], and agents targeting the pathway of bacterial fatty acid synthesis [78].

Fidaxomicin, a minimally absorbed macrocyclic antibiotic, as well as vancomycin, are both drugs of choice for CDI treatment, but the former has less disturbance in the microbiota and, thus, is associated with less recurrent rates compared to vancomycin [9,66,67,68,69]. In the mice model, fidaxomicin reduced the proportion of Clostridial growth to a lesser extent, but increased that of Bacteroidia, and it resulted in less disturbance in microbiota diversity [66]. The lesser impact on the microbiota composition of fidaxomicin, compared to vancomycin, promoted faster microbial recovery in the gut, which had more colonization resistance to *C. difficile*, and thus, compared with vancomycin, achieved a more sustained clinical cure after 30 days (OR 1.62, *p* = 0·030) and less recurrent rate (11% vs. 23%) after the end of treatment for CDI patients [67,68,69].

Ridinilazole, a narrow-spectrum antibiotic, with a non-inferior efficacy compared to vancomycin, had almost no disturbance in the gut microbiota, including several Bacteroidaceae and Clostridiaceae families, maintained the intestinal bile acid profile, and was associated with a lowered risk of recurrence in a phase 2 trial [70,71].

Cadazolid, a novel quinoxolidinone antibiotic developed for treating CDI, had non-inferiority for clinical cure compared to vancomycin in the phase 3 trial, with limited inhibition of the gut microflora, including the *B. fragilis* group and *Lactobacillus* spp. However, it is a pity, except for *Bifidobacterium* spp. [72,73].

Anti-sense antimicrobial agents, which means using the complementary binding of a modified anti-sense oligonucleotide (ASO) to a specific messenger ribonucleic acid (mRNA) in treating CDIs, has been investigated [74,75]. A group of 2′-O-methyl phosphorothioate gamer ASOs targeting five essential *C. difficile* genes achieved nanomolar minimum inhibitory concentrations for *C. difficile* [75]. The high specificity of the ASO for its target mRNA strengthens the binding of ASOs to its target mRNA, and improves specificity without offending microbiota [74,75].

The monoclonal antibody combination of actoxumab and bezlotoxumab, which bind to the receptor binding (known as combined repetitive oligopeptides (CROPs)) domains of *TcdA* and *TcdB*, respectively, was examined in treatment for primary or recurrent CDIs. Notably, because the recurrence rate was significantly lower in CDI patients treated with bezlotoxumab alone than those with the placebo in the phase 3 trial [79], the efficacies for reducing the recurrence rate by actoxumab-bezlotoxumab treatment might be only due to the effect of bezlotoxumab on facilitating normalization of the gut microbiota [77]. Another oral product delivered from the ovine polyclonal antibody and targeted toward *C. difficile* toxins successfully neutralized toxin production and did not interfere with the colonic microbiota in an in vivo hamster model and an in vitro human colon model, but its efficacies were still limited in pre-clinical models [76].

Enoyl-acyl carrier protein (ACP) reductase II (FabK) is a critical rate-limiting step within the synthesis pathway of FAS-II bacterial fatty acid, which supplies precursory component phospholipids found in bacterial cytoplasmic and spore-mediated membranes, as well as being essential in *C. difficile*. More importantly, it is structurally and mechanistically distinct from other isozymes found in gut microbiota species that make *C. difficile* FabK (CdFabK) an attractive narrow-spectrum target [78]. Though the FabK enzyme serves a potential role for the narrow-spectrum anti-*C. difficile* target, the related antimicrobial agents are still under investigation [78].

## 4. Experimental *C. difficile* Animal Models for Microbiota Analysis

To analyze the microbiota change during CDI, some germ-free or gnotobiotic animal models have been used [12,80,81,82]. Germ-free mice, which lack all microorganisms and allow for the transfer of selective bacterial species or whole fecal microbiota, serve as a completely blank microbial background to analyze the association of gut microbes with the host [80,82]. The transfer of human feces to germ-free mice has been applied to create a humanized gnotobiotic mouse mode since the 1980s, which was a revolutionary strategy to formulate in vivo systems of the human microbiota [81,83]. Inoculation of feces from a human donor into adult gnotobiotic recipient mice resulted in colonization by several strains from the donor that form an effective barrier against *C. difficile* [81,83]. Later, the use of rats implanted with human fecal microbiota were used as a model for studying the effects of diet on the human gut microflora [82,84]. Up to now, these rodents implanted with human microbiota have been considered the gold standard and cornerstone for establishing the causal role of microbiome alterations linked to human disease, with the advantages including: 1. they can be used to analyze the correlation between the phenotype and the environmental factors; 2. they can be used to establish a causal association between altered microbiomes and diseases of the human host; and 3. they represent a platform to apply integrated ‘‘omics’’ approaches to identify the causal components of altered microbiomes that drive disease [85,86].

Later, other improved animal models that are better mimics of human physiology, such as pigs and primates, were proposed [86]. Saul Tzipori et al. developed a reproducible piglet model for acute or chronic CDI with characteristic pseudomembranous colitis [87]. Presence of toxins in feces, body fluids, and serum, and a significant elevation of IL-8 in piglets with severe disease were noted, which suggested piglets to be a suitable animal model for investigating the role of virulence attributes, drug efficacy, and the evaluation of vaccine candidates [87]. Though pigs and primates are improved models, their greater cost can be constraining [86].

A critical challenge for developing animal models for microbiota analysis is the variances in the input microbiota from human donor feces, since there is a great influence of environmental factors on the differences in bacterial diversity and abundance in microbiota of donors [82]. Since not all humanized animal microbiota models can accurately represent different populations, selecting appropriate human fecal samples for the representative studies is still a task for microbiota analysis in animal models [82].

Another important issue in conducting animal models for microbiota analysis is the taxa that do colonize in animal models with human microbiota implants might differ substantially from those found in the human donors [86]. For example, though human and murine gut floras share 90% and 89% similarities in phyla and genera, respectively [88], the presence of unique microbes between humans and mice may pose a limitation on the generation of humanized gnotobiotic mouse models, particularly if these bacteria have host-specific physiological influences; for example, murine-segmented filamentous bacteria (SFB) [89]. Some host-dependent immune maturation might not occur after cross-species FMT, i.e., only mouse microbiota transplants, but neither those of humans nor rats, could induce immune cell expansions in germ-free mice [89]. Gut microbiota of mice generates higher concentrations of lactate than that of humans, although humans still produce higher levels of some short chain fatty acids (SCFAs), such as acetate and propionate [90]. So, in arranging microbiota studies using humanized gnotobiotic mouse models, selecting the appropriate mouse strains to reflect both the objectives of the study, the ability of the microbiota to colonize the gut, and the possible composition of microbiota and metabolite in mice gut after transplantation is an important issue in study design [82].

Regarding the preparation of human feces for implantation, the effect of temperature and atmospheric conditions on fresh samples varied depending on the donor, but storage for >24 h at 37 °C resulted in significant decline in the genera Lactobacillus, Enterococcus, Ruminococcus, and Eubacterium proliferated, whereas it differed for many Ruminococcaceae family members, such as Ruminococcus and Faecalibacterium [91]. So human fecal samples for FMT in mice should be fresh samples or samples prepared in maltodextrin-trehalose solutions stored at −80 °C before rapid thawing at 37 °C to preserve maximum flora resemblance [82,91].

## 5. Conclusions

To replenish the disturbed microbiota, the most efficient and commonly used method is FMT. FMT is listed as the alternative therapeutic choice for refractory or recurrent CDI in many guidelines. The drawback of FMT is the instable therapeutic effect because of the changing and diverse component of the donor’s microbiota. How to uniform the therapeutic effect of FMT is a major challenge now. Some commercialized liquid suspensions of donor microbiota, such as RBX2660, have been used. In the future, commercialized uniform donor microbiota might provide more efficient and more consistent therapeutic effects of FMT in treating CDI.

A mixture of probiotics, such as *Lactobacillus* species or *S. boulardii,* has been utilized extensively in preventing CDIs, and aims at less disturbance in the microbiota to prevent CDI recurrence after therapy cessation. Other than the preventive effect, it is a pity the evidence of the therapeutic effect of mixtures of probiotics in CDI is still limited. The disturbance on microbiota, from initial antibiotic exposure, subsequent infection by *C. difficile*, to finally treatment with metronidazole or vancomycin, could be expected to be very “severe” and a “disorder”. The limited therapeutic effect of mixtures of probiotics with only a few probiotic strains on this “severe” microbiota disturbance could be anticipated. Unless more potent probiotics mixtures are discovered, the effect of mixtures of probiotics in CDI remains a preventive instead of therapeutic role.

Numerous newly developed therapeutic agents aim at preserving microbiota during CDI treatment to prevent disease recurrence and might be useful in clinical patients with rCDIs. Less disruption in microbiota provides more chance for microbiota to recover and decrease the recurrent rate of CDI after treatment cessation.

It is a pity that, besides FMT, there are no better ways to replenish the disturbed microbiota up to now. Though some probiotics or drugs have been shown to increase some beneficial bacteria in the microbiota, these “treated” microbiota are still quite different from the “healthy” microbiota, especially in the bacterial diversity. Drugs that can replenish the disturbed microbiota, instead of using FMT, would be the target of new drug development in the future.

## Figures and Tables

**Figure 1 pathogens-10-00649-f001:**
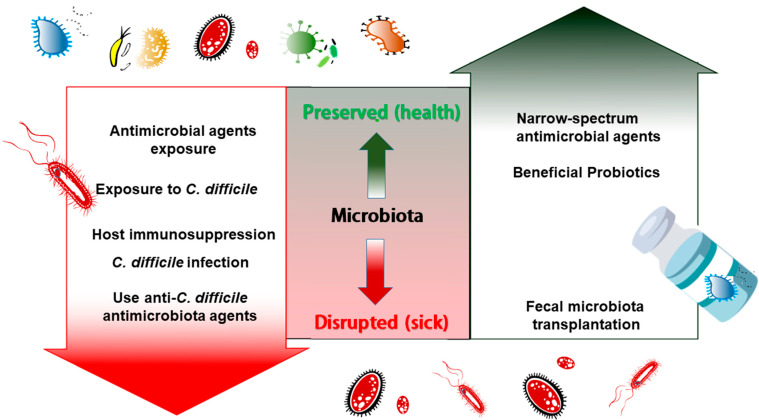
Fecal microbiota transplantation for *C. difficile* infection therapy.

**Table 1 pathogens-10-00649-t001:** Examples of influence on microbiota when treating adult patients with *Clostridioides difficile* infections (CDIs) with fecal microbiota transplantation (FMT) *.

Author	Year	Country	Recipient Numbers	Procedures of FMT	Effects on CDIs	Influence on Receipt Microbiota	Reference
Amy Langdon et al.	2013	USA	29	RBX2660, a liquid suspension of donor microbiota via enema	41.3% patients had no recurrence after a single dose; others had a recurrence and required a repeat dose	Reduced the abundance of antibiotic-resistant Enterobacteriaceae in two months	[34]
Yang Song et al.	2013	USA	14	Combined enteroscopy and colonoscopy	rCDI treated successfully by FMT	Members of Streptococci or Enterobacteriaceae were significantly decreased and putative butyrate producers, such as Lachnospiraceae and Ruminococcaceae, were significantly increased.	[30]
Sudhir K Dutta et al.	2014	USA	27	Combined enteroscopy and colonoscopy	All recipients had clinical resolution	Increased microbial diversity, increasing proportions of Lachnospiraceae (phylum Firmicutes) and reducing proportions of Enterobacteriaceae.	[29]
Anna M. Seekatz et al.	2014	USA	6	Colonoscopy	All recipients had clinical resolution of CDI following FMT and were recurrence-free up to six months.	Metabolites that increased following FMT were associated with bacteria classified within the Lachnospiraceae, Ruminococcaceae, and unclassified Clostridiales families.	[38]
Vijay Shankar et al.	2014	Finland	3	Colonoscopy	All patients had reduced diarrheal symptoms	Rich in members of *Blautia, Coprococcus*, and *Faecalibacterium.*	[26]
Michael Mintz et al.	2013–2016	USA	11	Colonoscopy	No rCDI after one year of follow-up	Reduced beta diversity differences between the donors and recipients, and increased in relative abundance of *F. prausnitzii.*	[16]
P C Konturek et al.	2014–2016	German	17	Colonoscopy	The healing rate of CDI was 94%. In all successfully treated patients no recurrent CDI was observed during follow-up (16 months)	Elevated abundance of beneficial bacterial species such as Lactobacillaceae, Ruminococcaceae, Desulfovibrionaceae, Sutterellaceae, and Porphyromonodacea after FMT.	[33]
Jonna Jalanka et al.	2016	Finland	14	The fecal suspension was infused into the cecum	The FMT treatment cleared rCDI from all patients	Dominated by Clostridium clusters IV and XIVa, with increased Bacteroidetes.	[24]
Braden Millan et al.	2016	Canada	20	Colonoscopy	FMT resulted in a resolution of symptoms	Decreased number and diversity of antibiotic resistance genes and increased Bacteroidetes and Firmicutes with reduced Proteobacteria.	[25]
Mohit Girotra et al.	2016	USA	29	Combined colonoscopy and nasojejunal tube	Marked improvement in all clinical parameters	Microbiota diversity increased with Proteobacteria decreased.	[28]
Christopher Staley et al.	2016	USA	39	Prepared as capsule and oral intake	Rapid resolution of rCDI symptoms	Taxa within the Firmicutes showed rapid increases in relative abundance and did not vary significantly over time. Bacteroidetes taxa only showed significant increases in abundance after one-month post-FMT.	[23]
Adrián Camacho-Ortiz et al.	2017	United Kingdom	7	Either colonoscopy or nasojejunal tube	Symptoms resolved in 57.1% patients after the first FMT and in 71.4% after the second dose.	The bacterial composition was dominated by Firmicutes, Bacteroidetes, and Proteobacteria at all-time points, and the microbiota were remarkably stable over time.	[22]
Z D Jiang et al.	2017	USA	72	Randomized to receive fresh, frozen or lyophilized FMT product via colonoscopy	Overall resolution of CDI was 87% during 2 months of follow-up.	Microbial diversity was reconstituted by day 7 with fresh or frozen product; by 30 days with lyophilized material.	[18]
Shaaz Fareed et al.	2017	USA	15	Either colonoscopy or nasojejunal tube	Prevented recurrent CDI for minimum of 3 months post-FMT in all patients.	Increased levels of Bacteroidetes and decreased levels of Proteobacteria	[21]
Jillian R-M Brown et al.	2018	Ireland	10	Both esophago-gastro-duodenoscopy and full colonoscopy	Nine out of ten patients improved clinically and remained *C. difficile* toxin negative for between 6 months and 2 years after a single FMT	FMT moves the microbiota of recipients towards that of the donor and improves bacterial diversity.	[31]
Zhi-Dong Jiang et al.	2018	USA	65	Encapsulated lyophilized fecal microbiota (*n* = 31) or frozen FMT (*n* = 34) by enema.	CDI recurrence rate after FMT: 84% in capsule group; 88% in FMT enema group, *p* = 0.76	Both products normalized fecal microbiota diversity while the lyophilized orally administered product was less effective in repleting Bacteroidia and Verrucomicrobia classes compared to frozen product via enema.	[20]
Christopher Staley et al.	2019	USA	18	encapsulated lyophilized fecal microbiota orally	All recovered clinically and were free of CDI	Members of the genera *Bacteroides, Parabacteroides*, and *Faecalibacterium* were positively correlated with donor similarity.	[19]
Hanne Jouhten et al.	2020	Finland	13	ND	Only recipients with rCDI successfully treated with FMT were included,	Specific donor-derived *bifidobacterium* can colonize rCDI patients for at least one year.	[27]

Abbreviations: CDI: *C. difficile* infection; rCDI: recurrent *C. difficile* infection; FMT: fecal microbiota transplantation; ND: no data; * Specific populations, such as inflammatory bowel disease and immunocompromised patients, were not listed.

## Data Availability

Data available in a publicly accessible repository.

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
