# Peer review of "Application of Microbiome Management in Therapy for Clostridioides difficile Infections: From Fecal Microbiota Transplantation to Probiotics to Microbiota-Preserving Antimicrobial Agents"

_pathogens, 2021, doi:10.3390/pathogens10060649_

Round 1

Reviewer 1 Report

The topic is very topical and the authors state and describe how different therapeutic approaches affect the microbiota of patients with C. difficile infection. There is a lack of critical thinking and suggestions on what to do and in which direction to go. The names of the bacteria are incorrectly stated throughout the text. Namely, "Bacteria - Italicize family, genus, species, and variety or subspecies. Begin family and genus with a capital letter. Kingdom, phylum, class, order, and suborder begin with a capital letter but are not italicized." Also, when the bacteria is first mentioned, they should be written in full and the abbreviation used later.
The conclusion and guidance as to what to do next or what the clinical study should look like should be supplemented.
Besides, a scheme  of the impact of different approaches to therapy on the microbiome would be desirable and would raise the quality of work.

Author Response

Dear the editor of Pathogens:

Enclosed, please find our revised manuscript entitled " Application of microbiome management in therapy for Clostridioides difficile infections: from fecal microbiota transplantation, probiotics to microbiota-preserving antimicrobial agents ". We have considered carefully for all concerns that were raised by the reviewers and have made the alterations clearly highlighted by using the "Track Changes" function in Microsoft Word in the tables of revised manuscript. We hope that our changes can clarify the points raised.

We look forward to hearing from you.

Best Regards,

Yuan-Pin Hung, MD

Department of Internal Medicine, Tainan Hospital, Ministry of Health and Welfare, Tainan, Taiwan

No 125. Zhong Shan Rd. Tainan. Taiwan

TEL: 886-6-2200055-3125

The topic is very topical and the authors state and describe how different therapeutic approaches affect the microbiota of patients with C. difficile infection. There is a lack of critical thinking and suggestions on what to do and in which direction to go.

Reply: After analyzing the current evidence about how different therapeutic approaches affect the microbiota of patients with CDI, the suggestions and directions had been added in the conclusion part (line 543-576). Thanks for the important suggestion.

The names of the bacteria are incorrectly stated throughout the text. Namely, "Bacteria - Italicize family, genus, species, and variety or subspecies. Begin family and genus with a capital letter. Kingdom, phylum, class, order, and suborder begin with a capital letter but are not italicized." Also, when the bacteria is first mentioned, they should be written in full and the abbreviation used later.

Reply: we had correct the names of the bacteria. We are sorry for the mistake.

The conclusion and guidance as to what to do next or what the clinical study should look like should be supplemented.

Reply: We had analyzed the current data about how different therapeutic approaches affect the microbiota of patients with CDI, the suggestions and directions had been added in the conclusion part (line 543-576).

Besides, a scheme of the impact of different approaches to therapy on the microbiome would be desirable and would raise the quality of work.

Reply: a scheme of the impact of different approaches to therapy on the microbiome had been added (line 84-87) and (figure).

Reviewer 2 Report

I have read with great interest the paper written by Chiu et al. on the current application of microbiota management in support of C. Difficile infection treatment. The authors refer to up-to-date research articles and the article has been well written and presented. 

However, I think the authors should give more focus to the current challenges in probiotics/prebiotics treatment (are they effective?what formulation?). Also, the authors should focus on the possible side effects of FMT (including death). Some of the answers of these question can be found in this recent research article (that should be cited) under the section called "current perspectives on GM-Based therapy" - DOI: 10.1152/ajpgi.00161.2019

Also, the authors should give some introductory information on gutmicrobiota (check these papers: 1) 10.1152/ajpgi.00161.2019 2) 10.3390/jcm9113705

Author Response

Dear the editor of Pathogens:

Enclosed, please find our revised manuscript entitled " Application of microbiome management in therapy for Clostridioides difficile infections: from fecal microbiota transplantation, probiotics to microbiota-preserving antimicrobial agents ". We have considered carefully for all concerns that were raised by the reviewers and have made the alterations clearly highlighted by using the "Track Changes" function in Microsoft Word in the tables of revised manuscript. We hope that our changes can clarify the points raised.

We look forward to hearing from you.

Best Regards,

Yuan-Pin Hung, MD

Department of Internal Medicine, Tainan Hospital, Ministry of Health and Welfare, Tainan, Taiwan

No 125. Zhong Shan Rd. Tainan. Taiwan

TEL: 886-6-2200055-3125

Email: mailto:[email protected]

I have read with great interest the paper written by Chiu et al. on the current application of microbiota management in support of C. Difficile infection treatment. The authors refer to up-to-date research articles and the article has been well written and presented. However, I think the authors should give more focus to the current challenges in probiotics/prebiotics treatment (are they effective?what formulation?).

Reply:

The information about the current challenges in probiotics/prebiotics treatment (including the effectiveness and the formulation) were added (line 386-409) and in conclusion part (line 552-564)

Also, the authors should focus on the possible side effects of FMT (including death).

Reply:

The possible side effects of FMT (including death) were added (line 301-306).

Some of the answers of these question can be found in this recent research article (that should be cited) under the section called "current perspectives on GM-Based therapy" - DOI: 10.1152/ajpgi.00161.2019

Reply:

I have read with great attention in the paper” A story of liver and gut microbes: how does the intestinal flora affect liver disease? A review of the literature”. It’s a well-written review article. Some information from this article had been added (line 61-87) and the article had been cited (Ref. 10).

Also, the authors should give some introductory information on gut microbiota (check these papers: 1) 10.1152/ajpgi.00161.2019 2) 10.3390/jcm9113705

Reply:

Both “A story of liver and gut microbes: how does the intestinal flora affect liver disease? A review of the literature” and “You Talking to Me? Says the Enteric Nervous System (ENS) to the Microbe. How Intestinal Microbes Interact with the ENS” are very good review article with many useful information about gut microbiota. Some information from this article had been added (line 61-87) and the article had been cited (Ref. 10 &11).

Reviewer 3 Report

Chiu et al. submitted their manuscript entitled “Application of microbiome management in therapy for Clostridioides difficile infections: from probiotics, fecal microbiota transplantation to microbiota-preserving antimicrobial agents” to Pathogens.

The recurrent infection with Clostridioides difficile (rCDI) has been successfully cured by fecal microbiota transplantation (FMT) from the health donor. As the title informed, the authors paid their attention also to other possible treatments as probiotics and anti-C. difficille agents.  Therefore, the most spread and known treatment within more than a decade is the FMT. I recommend changing the order in the manuscript to i) FMT, ii) probiotics and iii) anti-microbial agents.

It is known that there are differences between microbiota transplantation originated from adults and youngs as, e.g., Kim et al. in 2017 (Science) documented differentially established colonization resistance against S. Typhimurium in ex germ-free mice in their in vivo experiments with mice. It would be interesting to pay more attention to the influence of donors on FMT. Moreover, I miss entirely animal models of this illness. E.g., Tzipori´s group developed gnotobiotic piglet models of C. difficile infection and published several papers dealing with this topic.

Probiotic therapy of rCDI is not so spread as FMT. However, in contrast to FMT can be more easily and adequately defined, standardized and safer (see Khortus´ papers, especially the paper published this year). Please, search for it more deeply. It should be emphasized that it is possible to combine various probiotics with anti-C. difficile action, use a cross-feeding effect among probiotic bacterial strains, etc., to suppress C. difficile growth.

In the case of humans, the possibility of experimental work is limited, of course. As I wrote, it should not be forgotten on translational research with animal models. Thus, I recommend adding another part in the manuscript – iv) experimental C. difficile animal models and dedicate this part to animal models of this illness.

In reviews, it is almost author´s duty to present appropriate schema(s). I miss it. Please, support your ideas by schema, figure, etc., to make your manuscript/paper more instructive and attractive for readers.

At the end of my comments, I add some notices to the manuscript:

L20-21: If you do not wish to write "probiotic" only, I recommend "beneficial (probiotic) microorganisms".
L52-53: Please, use "beneficial" instead "nonpathogenic" if you are thinking about probiotic microorganisms.

L59: Please, use "microbiota" instead of "microflora".

L61:Microbiota instead of flora.

L61: E.g., "microorganisms" would be used instead of "agents".

L61: Lactobacillus should be in italics. You probably mean Lactobacillus rhamnosus GG. Please, complete it to be sure what you mean.

L64: Lactobacillus bifidus is old name for Bifidobacterium species. There is no relation between the old name and this review. Please, remove it.

Author Response

Dear the editor of Pathogens:

Enclosed, please find our revised manuscript entitled " Application of microbiome management in therapy for Clostridioides difficile infections: from fecal microbiota transplantation, probiotics to microbiota-preserving antimicrobial agents ". We have considered carefully for all concerns that were raised by the reviewers and have made the alterations clearly highlighted by using the "Track Changes" function in Microsoft Word in the tables of revised manuscript. We hope that our changes can clarify the points raised.

We look forward to hearing from you.

Best Regards,

Yuan-Pin Hung, MD

Department of Internal Medicine, Tainan Hospital, Ministry of Health and Welfare, Tainan, Taiwan

No 125. Zhong Shan Rd. Tainan. Taiwan

TEL: 886-6-2200055-3125

Email: mailto:[email protected]

Chiu et al. submitted their manuscript entitled “Application of microbiome management in therapy for Clostridioides difficile infections: from probiotics, fecal microbiota transplantation to microbiota-preserving antimicrobial agents” to Pathogens.

The recurrent infection with Clostridioides difficile (rCDI) has been successfully cured by fecal microbiota transplantation (FMT) from the health donor. As the title informed, the authors paid their attention also to other possible treatments as probiotics and anti-C. difficille agents.  Therefore, the most spread and known treatment within more than a decade is the FMT. I recommend changing the order in the manuscript to i) FMT, ii) probiotics and iii) anti-microbial agents.

Reply:

We change the order of the topics as suggestion: i) FMT, ii) probiotics and iii) anti-microbial agents and also change the topic: Application of microbiome management in therapy for Clostridioides difficile infections: from fecal microbiota transplantation, probiotics, to microbiota-preserving antimicrobial agents.

It is known that there are differences between microbiota transplantation originated from adults and youngs as, e.g., Kim et al. in 2017 (Science) documented differentially established colonization resistance against S. Typhimurium in ex germ-free mice in their in vivo experiments with mice.

Reply:

The article “Neonatal acquisition of Clostridia species protects against colonization by bacterial pathogens” by Kim et al. was a very interesting article, and we had discuss it (line 61-87) and cite this article (Ref.12).

It would be interesting to pay more attention to the influence of donors on FMT.

Reply:

The influence of donors on FMT had been discussed in clinical aspect (line 183-197) and in mice model design (line 505-511).

Moreover, I miss entirely animal models of this illness. E.g., Tzipori´s group developed gnotobiotic piglet models of C. difficile infection and published several papers dealing with this topic.

Reply:

A section discussing the animal model with the article “Piglet Models for Acute or Chronic Clostridium difficile Illness” by Saul Tzipori et al. had been added (line 474-540).

Probiotic therapy of rCDI is not so spread as FMT. However, in contrast to FMT can be more easily and adequately defined, standardized and safer (see Khortus´ papers, especially the paper published this year).

Reply

We had consult the articles by Alexander Khoruts (line 392-409) and cite the articles (Ref 64 & 65).

Please, search for it more deeply. It should be emphasized that it is possible to combine various probiotics with anti-C. difficile action, use a cross-feeding effect among probiotic bacterial strains, etc., to suppress C. difficile growth.

Reply:

The effect of “combine various probiotics” (the mixture of probiotics) was illustrated (379-386) and discussed (552-564).

In the case of humans, the possibility of experimental work is limited, of course. As I wrote, it should not be forgotten on translational research with animal models. Thus, I recommend adding another part in the manuscript – iv) experimental C. difficile animal models and dedicate this part to animal models of this illness.

Reply:

We add the section: iv) experimental C. difficile animal models (line 474-540) to dedicate the animal model of microbiota.

In reviews, it is almost author´s duty to present appropriate schema(s). I miss it. Please, support your ideas by schema, figure, etc., to make your manuscript/paper more instructive and attractive for readers.

Reply:

We add the schema of our idea (Figure).

At the end of my comments, I add some notices to the manuscript:

L20-21: If you do not wish to write "probiotic" only, I recommend "beneficial (probiotic) microorganisms".

Reply:

We revised it as suggestion (25-26).

L52-53: Please, use "beneficial" instead "nonpathogenic" if you are thinking about probiotic microorganisms.

Reply:

We use "beneficial" instead "nonpathogenic" (line 52-53).

L59: Please, use "microbiota" instead of "microflora".

Reply:

We use "microbiota" instead of "microflora" (line 75).

L61:Microbiota instead of flora.

Reply:

We use Microbiota instead of flora (line 77).

L61: E.g., "microorganisms" would be used instead of "agents".

Reply:

We use "microorganisms" would be used instead of "agents" (line 320).

L61: Lactobacillus should be in italics. You probably mean Lactobacillus rhamnosus GG. Please, complete it to be sure what you mean.

Reply:

We revised it as Lactobacillus rhamnosus GG (line 320).

L64: Lactobacillus bifidus is old name for Bifidobacterium species. There is no relation between the old name and this review. Please, remove it.

Reply:

We removed  Lactobacillus bifidus (line 323)

Round 2

Reviewer 1 Report

Although I suggested minor changes the work looks much better now and the authors have accepted the suggestions and improved the review article. I have no further comments.

Reviewer 3 Report

The changes in the manuscript are so far-reaching that the manuscript should not be submitted as version 2 but as a quite new submission of a new manuscript.